# A Mini Review of Physicochemical Properties of Starch and Flour by Using Hydrothermal Treatment

**DOI:** 10.3390/polym14245447

**Published:** 2022-12-13

**Authors:** Edy Subroto, Rossi Indiarto, Vira Putri Yarlina, Afifah Nurul Izzati

**Affiliations:** Department of Food Industrial Technology, Faculty of Agro-Industrial Technology, Universitas Padjadjaran, Bandung 45363, Indonesia

**Keywords:** annealing, flour, hydrothermal, heat moisture treatment, starch

## Abstract

Starch and flour from various plants have been widely used for sundry applications, especially in the food and chemical industries. However, native starch and flour have several weaknesses, especially in functional, pasting, and physicochemical properties. The quality of native starch and flour can be improved by a modification process. The type of modification that is safe, easy, and efficient is physical modification using hydrothermal treatment techniques, including heat moisture treatment (HMT) and annealing (ANN). This review discusses the hydrothermal modifications of starch and flour, especially from various tubers and cereals. The discussion is mainly on its effect on five parameters, namely functional properties, morphology, pasting properties, crystallinity, and thermal properties. Modification of HMT and ANN, in general, can improve the functional properties, causing cracking of the granule surface, stable viscosity to heat, increasing crystallinity, and increasing gelatinization temperature. However, some modifications of starch and flour by HMT and ANN had no effect on several parameters or even had the opposite effect. The summary of the various studies reviewed can be a reference for the development of hydrothermal-modified starch and flour applications for various industries.

## 1. Introduction

Natural flours and starches from various plants generally have several weaknesses in their characteristics, which limit their application in various industries [1,2,3]. These characteristics can be improved through physical, enzymatic, and chemical modification processes [4,5,6,7,8]. Various types of modifications can be applied, including physical modification by hydrothermal treatment. Hydrothermal treatment has several advantages compared to other treatments: greater safety and environmentally friendliness, as it does not use harmful chemicals; greater ease of use, as process control is needed only on temperature, moisture content, and time settings; lower costs, as the materials and equipment are simple; and greater effectiveness in improving the properties of starch and flour from various plants [9,10,11].

Other starch treatments have several advantages but also some disadvantages. Chemical modification can produce changes in starch molecules quickly and significantly but has the potential to leave chemical residues and is not environmentally friendly. Enzymatic modifications also result in rapid, specific, and significant changes in starch molecules, but controlling the process is relatively difficult [3,12,13,14]. Hydrothermal modification is known to be effective in modifying starch. However, the treatment applied must be appropriate so that the level of change or improvement in characteristics is as desired [15,16,17,18,19,20,21,22,23]. Therefore, it is necessary to select the right or optimal modification method and process conditions to obtain good starch quality. The most frequently used hydrothermal modifications are annealing (ANN) and heat moisture treatment (HMT), because these two types of physical modifications have proven effective in improving the characteristics of various types of starch and flour, especially in relation to their stability in heat and under acid conditions [14,24,25]. Several studies related to the HMT of sweet potato starch [26] and the ANN of sweet potato starch [27] were able to have a significant effect on the pasting properties of starch, which indirectly also increased heat stability.

Hydrothermal modifications (ANN and HMT) are the type of physical modification that has the principle of using specific humidity, heat, and pressure levels [14,16,28,29,30]. Modification of ANN took place under conditions of excess (>60% *w*/*w*) or moderate (40% *w*/*w*) water content at temperatures above the glass transition temperature (T_g_) and below the initial gelatinization temperature (To), i.e., ±40–75 °C for a certain period [31,32,33,34,35]. The HMT modification took place at a limited water content (< 35%), a temperature above the T_g_ and above the starch gelatinization temperature, namely ± 84–120 °C for 15 min–16 h [14,36,37].

The positive impact of hydrothermal modification is an opportunity for the development of starch and flour-based products, especially for various food and chemical industries [38,39,40]. Therefore, a review of the modification of flour and starch by hydrothermal treatments (ANN and HMT) is needed as a basis for further development so that its application is wider and more efficient. The main objective of this review is to obtain a comprehensive comparison of the functional, pasting, and physicochemical properties of hydrothermally modified starch and flour (ANN and HMT) from various plant sources so that they can be used as a reference for the development of modified starch and flour for various industrial needs. The characteristics of ANN- and HMT-modified starch were assessed from several parameters, including functional properties, morphology, pasting, crystallinity, and thermal properties. The following are characteristics of hydrothermally modified starch and flour.

This review was carried out by collecting various articles using the Scopus, Web of Science, and Google search engines, using keywords of hydrothermal treatment, heat moisture treatment, annealing, from August 2021 to October 2022. The articles obtained were selected based on their suitability and were not included in the predatory journals.

## 2. Functional Properties of Hydrothermal Modified Starch and Flour

The functional properties of ANN and HMT hydrothermal-modified starch and flour are quite diverse, and they can be in the form of increasing or decreasing the parameter values. However, in general, both experienced a decrease in swelling power (SP) and solubility. The decrease in swelling power and solubility was related to the changes in the overall arrangement of starch granules. There was an improvement in the structure of amylopectin, amylose, and amylopectin levels in starch granules where swelling power increased with high amylose content and the intensity of intra- and interchain interactions [41,42,43]. The molecular structure of amylose and amylopectin can be seen in Figure 1.

Generally, HMT has a greater effect on the solubility and crystallinity of flour than starch. Sun et al. [44] reported that HMT in sorghum flour decreased swelling volume and solubility, which was greater than that in sorghum starch. HMT also increased the crystallinity of sorghum flour, which was greater than that of sorghum starch. However, the effect can be different in certain starches, which are affected by the amylose-amylopectin composition in starch and non-starch components in flours. Non-starch components such as proteins, phenolic compounds, lipids, and minerals can form complexes with starch during treatment, which decreases solubility and increases crystallinity [44,45]. Amylose-amylose, amylose-amylopectin, and/or amylopectin-amylopectin, decreased hydration and diffusion, leading to increased molecular mobility and decreased hydroxyl groups, formation of amylose-lipid complexes, decreased amylose leaching, temperature, and humidity [14,26,28,31,35,37,46]. Several studies related to the effect of hydrothermal modifications on the functional properties of starch and flour can be seen in Table 1.

Hydrothermal treatments (HMT and ANN) generally cause a decrease in swelling power and solubility in line with water absorption capacity and inversely with an increase in gel hardness. The less water absorbed by the starch molecules, the greater the starch gel hardness due to structural damage and cross-link interactions between starch chains (especially the stronger amylose fraction), making the gel stiffer [16,22,65]. In addition, the increase in gel hardness was affected by the formation of a wider junction zone [35,36]. The decrease in functional properties of HMT-modified starch was relatively more significant than in ANN. This is because HMT tends to limit the movement of water molecules in starch granules as their crystallinity increases [60,66,67,68].

The decrease in swelling power and solubility values was shown by the modification of HMT [20,25,28,40,44,47,48,50,52,53,55,61,62] and was cyclical [26,36,56]. Modification of HMT causes rearrangements in starch granules, increasing the mobility of starch molecules, which results in increased bond interactions between amylose-amylose and amylose-amylopectin, formation of amylose-lipid molecular complexes, changes in diffusion and hydration mechanisms [69,70,71]. These changes are due to the reduction of hydroxyl groups involved and available, which causes a decrease in the value of swelling power, solubility, and amylose leaching as well as an increase in starch gel hardness [14,72,73].

Changes in functional properties can also be affected by the applied cycle of hydrothermal treatments. As the modification cycle increases, the level of interaction between the starch chains also increases [74,75]. This event is indicated by a decrease in the value of swelling power and solubility. However, there is a threshold change in certain starch modification cycles where cycles 4 and 5 in sweet potato starch did not show significant changes [26]. Although the cycle has an impact on the functional properties, in general, the decrease is more affected by temperature and humidity than by the number of cycles applied, because the addition of the HMT cycle does not cause the amylopectin crystals to break. However, high temperature is more influential, which then has an impact in the form of a decrease in the degree of starch crystallinity [36,76].

The decrease in swelling power and solubility was shown by ANN modifications and cyclical ANN. The modification of ANN made the crystalline structure stronger as the hydration level of the amorphous portion of the starch decreased [31,32]. This is indicated by a decrease in swelling power and solubility as well as an increase in gel hardness due to a decrease in gel volume [14,65,77].

Changes in functional characteristics can also be affected by the applied cycle. Cyclic ANN modifications in sweet potato starch [27] and red adzuki bean starch [59] showed a decrease in swelling power and solubility with cycles and when ANN was applied. In addition to cycles, modifications can also be made continuously. However, cycle modification is more effective than continuous modification, which is represented by the lower swelling power and solubility values in the cycle modification [34,35]. This is due to starch degradation events that may occur during the cooling/transition phase of the modified cycle treatment. This phase causes the number of small starch granules to increase; the natural structure of starch is damaged, the double helix structure which was originally damaged undergoes partial repair, and redistribution of water molecules occurs in the amylose and amylopectin active sites along with an increase in the interaction, mainly through hydrogen bonds, which then increases the interaction between the amylose-amylopectin chains in starch [15,22,36]. The lower value of swelling and diffusion capacity of amylose in cyclically modified starch compared to continuous indicates that starch has a more stable structure [26,57]. However, in general, the decrease in functional parameters that occurs is more affected by temperature and humidity than by the number of cycles applied [36].

Changes in the functional properties of starch will be indirectly related and reflected in the characteristics of the granules that can be identified through microscopic observations, which are known as the morphological properties of starch.

## 3. Morphological Properties of Hydrothermal Modified Starch and Flour Granules

Morphological properties of ANN- and HMT-modified starch granules showed a change, but some studies were not significant. Changes are seen in the size and irregularity of the starch granule surface: texture and the presence of holes, fissures, cavities, indentations, or cracks [23,78,79]. These changes are largely determined by internal factors (amylose and amylopectin components) and external factors (temperature, moisture, time, and environmental pressure). These two factors influence each other in the process of aggregation and agglomeration during the hydrothermal modification process. Starch with high–medium amylose content produced more aggregated starch granules with a more irregular surface, whereas starch with a low amylose content decreased the integrity of starch granules, which caused starch granules to be more susceptible to partial gelatinization and physical changes [14,17,26,37,52,56,57,60]. Several studies related to the effect of hydrothermal modifications on the morphology of starch and flour can be seen in Table 2.

Changes that occur in granule morphology tend to have a more significant effect on HMT-modified starch than ANN, which is proven to be dominant in combination modification [14,83]. This change is also in line with the birefringence pattern observed through analysis with SEM, LM, PLM, and CLSM. Granule morphology of several types of starch/flour modified by HMT and ANN analyzed by scanning electron microscope (SEM) can be seen in Figure 2. ANN- and HMT-modified starch showed different birefringence patterns. This difference reflects the different effects on the internal conditions of starch granules where the radial orientation of the helical structure and its mobility change according to the heat energy obtained during modification [57,68]. The birefringence pattern of the starch granules after hydrothermal treatment can be analyzed by a polarized light microscope (PLM) which can be seen in Figure 3, or by using a light microscope (LM) and confocal laser scanning microscope (CLSM), as in the example that can be seen in Figure 4. Modification of ANN relatively maintains the birefringence pattern with the characteristic of a clear and bright granule pattern without any change in polarization cross, while HMT causes a decrease in brightness intensity with a more opaque appearance of the starch hilum structure (growth rings are more difficult to detect) [84,85]. The insignificant changes or no changes in the birefringence pattern indicated that the starch granule structure was more stable, while the changes in the form of a decrease occurred due to the destruction of amylose-amylose and amylose-amylopectin chains during modification [27,35,56].

HMT modifications have different effects on starch and flour granules for each type of starch and treatment. Several factors affect the morphology of starch granules during HMT, including internal factors (amylose and amylopectin composition), as well as conditions before and during modification (temperature, humidity, time, and pressure) [87,88]. These factors trigger an increase in starch polymer chain interactions to associate or strengthen bonds between amylose-amylopectin chains, which are referred to as aggregation and agglomeration processes [26,56].

Aggregation and agglomeration processes take place along with swelling and partial gelatinization during modification. These events cause morphological changes in the form of indentations, cracks, holes/gaps/cavities, irregularities, and changes in the size/shape of starch granules, so that the amorphous part of the starch granules becomes more compact [18,79,89]. However, the opposite effect is also possible. This is due to the components that make up the amorphous part (amylose and amylopectin) in each type of starch being different. Starch with medium–high amylose content tends to produce more aggregated starch granules with a more irregular surface, while starch with low amylose content shows a decrease in the integrity of starch granules so that they are more susceptible to partial gelatinization and physical changes [14,17,26,37,52,56,57,60].

The agglomeration process can also continue along with HMT modification cycles and times which cause disruption of starch granules [26]. This is related to the birefringence properties of starch granules. The birefringence pattern shows a change marked by the loss of a polarization cross over time and the number of modification cycles. The cross-polarization of starch is related to the event of partial gelatinization and changes that occur in the starch granules, such as holes in the center of the granules formed during agglomeration. A polarization cross can be represented by a wide dark area in the middle of the cross [28,40,48].

HMT modification also causes a decrease in brightness intensity; the starch hilum becomes darker, the edges of the starch granules become more opaque, and the growth rings are more difficult to detect. These changes also indicated a decrease in the starch crystallinity index. This decrease occurs due to the destruction of amylose-amylose and amylose-amylopectin chains during cyclical and continuous modifications. [56]. This result is also in accordance with the study of Zhao et al. [57] on mung bean starch, which resulted in a decrease in the starch crystallinity index. However, several other studies reported an increase in birefringence patterns in starch after HMT modification. Changes in the birefringence pattern indicate changes that occur in the starch granules are characterized by the loss of the radial orientation of the helical structure that makes up the starch granules. This change occurs due to the mobility of the helical structure due to heat energy during modification [40,48].

Almost the same thing happened to the modification of starch and flour by ANN. ANN modifications in potato, barley, lentil, oat, and wheat starch did not show significant morphological changes, but other high-amylose starches showed significant changes [31,46,90]. The changes that occurred were reported by Molavi et al. [17] in the form of surface irregularities of starch granules, including their size and shape. Other changes in the form of cracks or the appearance of pores on the starch surface over time and modification cycles are due to amylose leaching during modification [27].

The effect of ANN modification on the morphological characteristics of starch can also be seen from its birefringence pattern. The relative ANN modification maintains the birefringence pattern. The results of the study that did not show changes in the birefringence pattern were found in ANN modifications for corn starch, lentils, and peas [40], cyclical and continuous ANN modification on red adzuki bean starch [59], and ANN modification on sweet potato starch [27]. The results of these studies show that the clear birefringence or Maltese cross pattern is displayed clearly and brightly and the polarization cross remains in the center of the particle. This indicates that no significant changes were observed [27]. ANN modification can strengthen the crystal structure, encourage the process of rearranging starch molecules, and increase the overall stability of starch. There were no significant changes in the birefringence and polarization cross patterns, indicating that the starch granules were not gelatinized during ANN modification due to their more stable structure. ANN modification can maintain the internal integrity of starch granules, and cycle treatment is more effective than continuous treatment in modifying starch characteristics [27,56,59].

However, if other studies found different birefringence patterns, this indicates that the changes that occur in the starch granules are not always the same due to various factors, both internal and external, during different processes [14,87,88]. Changes that occur in granule morphology and functional properties can be further identified by observing other parameters, namely pasting properties. Pasting properties is a mini simulation of food processing that is useful for determining the stability of materials.

## 4. Pasting Properties of Hydrothermal Modified Starch and Flour

The pasting properties of various ANN- and HMT-modified starches showed very different results on the same parameters, either with or without the application of certain cycles. In general, hydrothermal modification causes a decrease in peak viscosity (PV) and breakdown (BD) [17,91,92]. The decrease in PV and BD occurred due to the improvement of the molecular arrangement of the starch granules, especially the crystalline region, which was influenced by the increase in the intra- and inter-starch granule interactions. The decrease in PV and BD can also be influenced by the presence of carboxyl groups which then affect the structure and interactions between amylose-amylopectin [93,94,95]. This event is also closely related to other processes, including a decrease in the hydration level of the amorphous part, the expansion process, solubility, and amylose leaching, as well as an increase in the thermal stability of starch. This event is in line with the increase in hot paste viscosity or through viscosity (TV) which is commonly found in hydrothermal modifications. The increase in TV is an indication that the risk of separating the starch granule structure is getting smaller (the starch structure is getting stronger) [26,27,31,96]. In addition to the TV value, hydrothermal modification generally causes an increase in the final viscosity (FV) and setback viscosity (SB) values. The increase in FV and SB indicated that the cold stability of starch was not good. This change was caused by the aggregation and restructuring of amylose molecules during the cooling phase, which continued to increase with time and modification cycles. Increased TV and FV values are generally found in the hydrothermal modification of ANN [17,27,34,60]. Several studies related to the effect of hydrothermal modifications on the pasting properties of starch and flour can be seen in Table 3.

Viscosity changes that occur are more significant in HMT-modified starch than in ANN [2,14,25,40,82,86]. Changes in viscosity can be seen from the pasting properties graph, which shows an improvement in the heat stability of ANN- and HMT-modified starch with differences in the gelatinization pattern. The gelatinization pattern of ANN-modified starch tends to show a type B gelatinization pattern, while the HMT-modified starch tends to show a type C gelatinization pattern [14,21,24,98]. Gelatinization profile B is starch with moderate swelling power (moderate) and similar characteristics to profile A (lower PV value of profile B). Gelatinization profile C is starch with limited expansion, which is characterized by the absence of PV and BD with relatively constant or increasing viscosity during food processing [3,10,14,60].

Apart from the changes in pasting properties, HMT and ANN showed better efficiency when applied with certain cycles rather than singly. Treatment cycles (a certain number of cycles) were more effective than single or continuous treatment in modifying starch [11,26,56,57,99]. The hydrothermal modifications that have a significant effect are HMT modifications rather than ANN and other combination modifications [27,34,63,82].

Changes in the pasting properties of starch due to modification are strongly influenced by the process of perfecting the crystalline part that occurs. This event can be studied further through an approach to starch crystallinity, which can be analyzed using X-ray diffraction (XRD) [63,100].

## 5. The Crystallinity of Hydrothermal Modified Starch and Flour

Starch crystallinity can be seen from the relative crystallinity (RC) index with an RC value of 15–45%. Starch crystallinity does not always show the same results because there are varying influencing factors, including the nature of the natural polymorph of the material, which includes amylose/amylopectin ratio, amylopectin chain length and branching, the mole percentage of amylopectin fraction/degree of polymerization (DP), crystal size, and interactions, and a constituent double helix (density and orientation) [14,47,101,102]. Several studies related to the effect of hydrothermal modifications on the crystallinity of starch and flour can be seen in Table 4.

The degree of crystallinity greatly determines the type of starch crystallinity, which is divided into three types, namely types A, B, and C. The type of starch crystallinity (A-type, B-type, and C-type), as shown by X-ray diffraction patterns, can be seen in Figure 5. Crystalline type A is generally a cereal group; type B is found in tubers, fruits, and high amylose starch), while type C is a combination of types A and B, which are found in legume starch. In addition to these three types, there is also type V, which is a group of amylose that has complex bonds with other organic compounds [14,101,104].

Starch crystallinity index may increase, decrease, or change insignificantly with the main characteristic of hydrothermally modified starch, namely the type of crystallinity that is relatively fixed or transformed into type A [15,18,82,86,103,106,107]. The crystallinity, as shown by X-ray diffraction patterns of modified starch/flour by ANN and HMT, can be seen in Figure 6. Although the polymorph type between ANN and HMT starch is the same, ANN-modified starch has higher crystallinity, better than HMT and the combination. This is evident from RC_ANN_ > RC_Natural/HMT/Combination_. Optimum ANN modification can produce an orderly and perfect structure. The choice of the type of modification affected the crystallinity of starch more significantly than the choice of treatment (cycles/continuous), although cycles/continuous treatments still had an effect on starch [35,108]. Characteristics of starch crystallinity are quite diverse, but hydrothermally modified starch ANN and HMT tend to have one thing in common. In addition to crystallinity, starch characteristics can also be identified based on thermal properties analyzed by a differential scanning calorimeter (DSC).

## 6. Thermal Properties of Hydrothermal Modified Starch and Flour

Thermal properties in this study focused on three parameters, namely gelatinization temperature, gelatinization temperature range (GTR), and heating enthalpy (ΔH). The characteristics of the three parameters from each study are not necessarily the same, even though the modified method used is the same. However, in general, ANN and HMT have similar thermal properties in the form of increasing gelatinization temperature [2,14,68,109,110]. The increase in gelatinization temperature occurs because the bonds and interactions in the starch granules are getting stronger, making the starch conformation more stable (increased heat stability). The more stable the starch, the higher the initial gelatinization temperature [14,65,111]. Although the gelatinization temperature of both methods increased, other parameters showed different and quite varied results. Several studies related to the effect of hydrothermal modifications on the thermal properties of starch and flour can be seen in Table 5.

The heating enthalpy parameter (ΔH) of ANN-modified starch could be an increase, decrease, or insignificant change, while HMT-modified starch generally decreased [13,68,113,114]. The increase in ΔH indicates the amylose-amylose and/or amylose-amylopectin interactions that occur, causing the formation of new double helix bonds. The decrease in ΔH indicates the process of melting and expansion of starch granules accompanied by partial melting of the crystalline part that occurs when the modification temperature approaches the initial temperature of natural starch gelatinization [13,14]. The insignificant change in ΔH indicates that the arrangement of the double helix bonds during the refinement of the crystalline section has no significant effect, so the energy requirements during the process also do not change significantly [31,108].

Likewise, the gelatinization temperature range (GTR) of ANN- and HMT-modified starch showed different results. The GTR of ANN-modified starch showed a decrease, while the HMT modification caused an increase in GTR. GTR is a parameter of the homogeneity of the starch granule structure. A decrease in GTR indicates a more homogeneous starch granule structure, and more even crystal melting occurs, while an increase in GTR indicates a more heterogeneous starch granule structure. The increase in GTR is also in line with the increase in gelatinization temperature. Both indicate an increase in the need for heat/temperature energy for the melting process [31,35,40,47,50,57,60,106,108]. If the hydrothermal modification takes place at optimum conditions, the characteristics of starch can be improved by the characteristic of increasing the gelatinization temperature of starch (the crystalline structure which was initially weak becomes more stable) with or without decreasing H and the starch crystallinity index. This can be achieved by selecting the right type of modification. One example is the modification of ANN, which can produce better stability of starch configuration [10]. In addition, cycles also affect the characteristics of the results. The modified starch of HMT and ANN cycles had a starch crystal structure that was more perfect than the single or continuous treatment. This is evidenced by the higher gelatinization temperature and lower ∆H [27,31,56].

## 7. Applications of Hydrothermal Treated Starch and Flour

Hydrothermal-treated starch and flour (ANN and HMT) can be applied to a variety of food products, especially those that require a stable product when exposed to heating/cooling and a strong texture. HMT- and ANN-modified starches can be used in noodles, baking products, frozen foods, canned foods, edible films, and other various products [14,38,115,116]. The use of HMT- and ANN-starch in noodle production can improve firm texture and cooking quality [65,117,118]. HMT-flour/starch is able to maintain the quality of dough and bread properties produced compared to untreated flour [65,117,118]. HMT-starch can increase the tensile strength, elongation, and thickness of edible films [119,120]. HMT-starch is also able to maintain the stability of the viscosity of tomato sauce during heating [38]. Some applications of hydrothermal-treated starch and flour (HMT and ANN) can be seen in Table 6.

## 8. Conclusions and Future Research

Hydrothermal treatments (ANN and HMT) in general have a significant effect on the physical and functional properties of starch, granule morphology, pasting properties, crystallinity, and the thermal properties of starch and flour. However, several studies have produced various characteristics depending on the source of starch/flour and the conditions of the modification process. The combination of hydrothermal modification with ANN and treatment cycles can increase the effectiveness and quality of modified starch products so that they are potentially suitable for application in various industries in the development of flour and starch from various plant sources.

Hydrothermal-modified starch and flour have been widely used for various applications in the food, chemical, and pharmaceutical fields. However, modification of starch through hydrothermal treatments still has several weaknesses, such as instability in some physical properties, the appearance of cracks or wrinkles on the surface of the starch granules, and a darker color. Therefore, several additional treatments are needed to improve these characteristics. Further studies are needed to combine with other treatments, especially environmentally friendly ones, to improve the characteristics. These treatments include oxidation with ozonation, the use of vacuum pressure to reduce heat, cavitation energy through ultrasonication, irradiation with ultraviolet (UV) light, sublimation with freeze drying, and other treatments. Oxidation by ozonation can improve color brightness and solubility; vacuum pressure can improve process efficiency; ultrasonication can increase the porosity of the granule surface and improve process efficiency; UV light can increase solubility; and sublimation can increase the porosity of the granules and increase the solubility. In addition, the effects of several hydrothermal treatments can also produce porous starch granules, so that they have the potential to be developed into porous starch which can be used as an adsorbent, encapsulant, and fast-cooking starch.

## Figures and Tables

**Figure 1 polymers-14-05447-f001:**
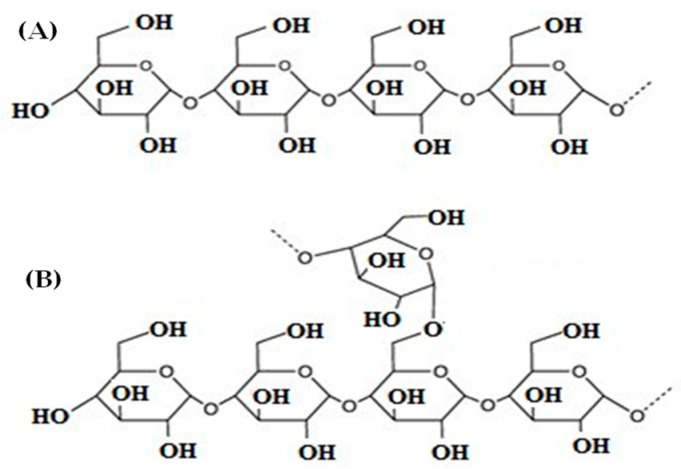
The molecular structure of amylose (**A**) and amylopectin (**B**).

**Figure 2 polymers-14-05447-f002:**
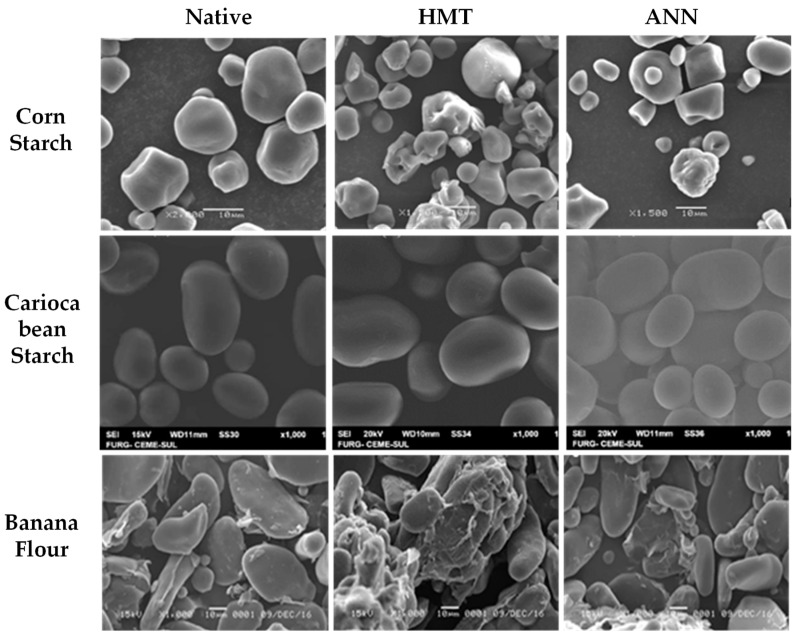
Starch granule morphology of HMT and ANN treatment was analyzed by scanning electron microscope (SEM) of maize starch [86], with permission from John Wiley and Sons, 2016, carioca bean starch [82], with permission from Wiley and Sons, 2018, and banana flour [63], with permission from Elsevier, 2019.

**Figure 3 polymers-14-05447-f003:**
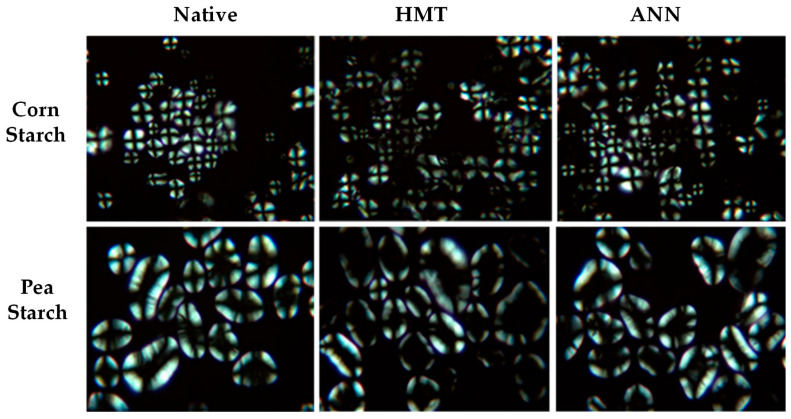
The birefringence pattern of starch granules modified by HMT and ANN was analyzed by a polarized light microscope (PLM) of corn and pea starches at the magnification (400×) [40], with permission from Elsevier, 2009.

**Figure 4 polymers-14-05447-f004:**
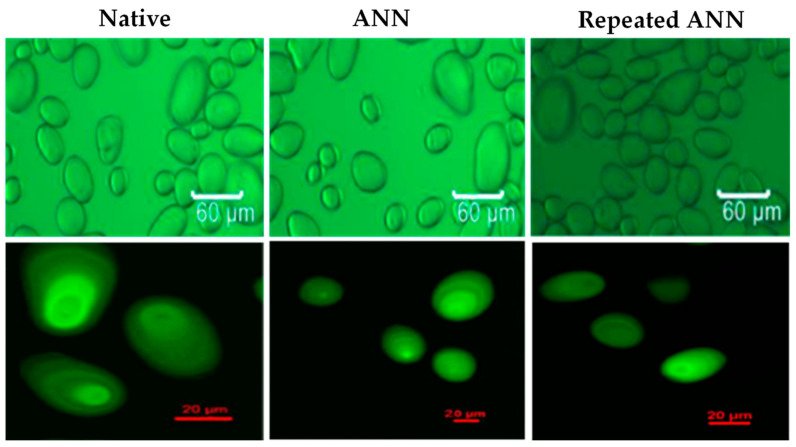
Starch granule morphology after hydrothermal treatment was analyzed by light microscope (LM) (**above**) and confocal laser scanning microscope (CLSM) (**below**) of potato starch [35], with permission from Elsevier, 2018.

**Figure 5 polymers-14-05447-f005:**
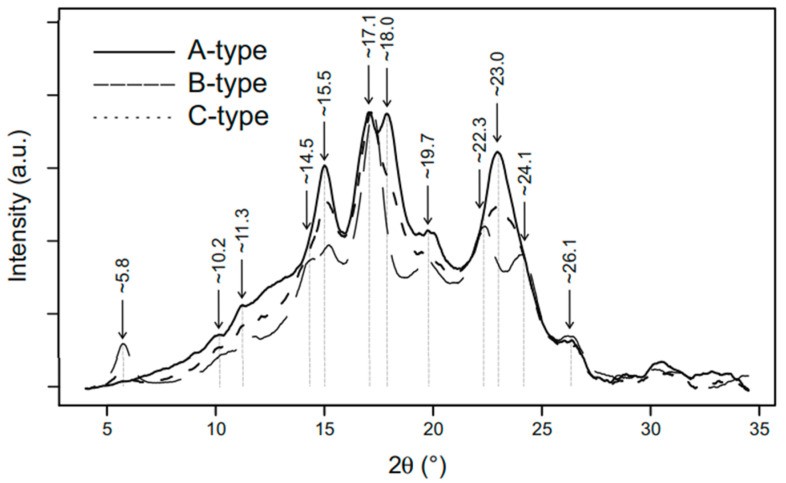
The type of starch crystallinity (type A, type B, and type C) as shown by X-ray diffraction patterns [105], with permission from Springer Nature, 2018.

**Figure 6 polymers-14-05447-f006:**
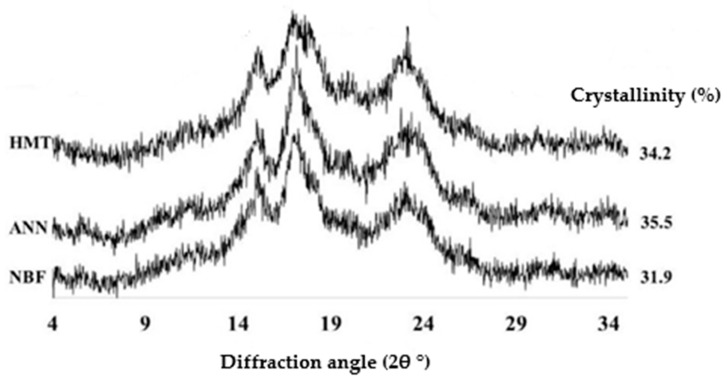
XRD patterns of native banana flour (NBF), annealing (ANN), and heat moisture treatment (HMT) [63], with permission from Elsevier, 2019.

**Table 1 polymers-14-05447-t001:** Several studies related to the effect of hydrothermal modifications on the functional properties of starch and flour.

Modification Types	Source of Starch/Flour	Effect on Functional Properties	References
HMT (100 °C, 10 h, 30% moisture)	Potato, taro, new cocoyam, true yam, cassava	Decreased swelling power	[47]
HMT (110 °C, 1 h, 15%, 20%, 25% moisture)	Rice	Decreased swelling power and solubility	[28]
HMT (120 °C, 12 h, 15%, 20%, 25%, 30%, and 35% moisture)	Mung beans	Decreased swelling power but increased solubility	[48]
HMT (90, 110, 130 °C; 1, 8, 15 h; 18, 23, 28% moisture)	Sweet potato flour	Decreased swelling power	[49]
HMT (120 °C, 3 h, 15, 20, 25% moisture, pH 5.0 or 6.0)	Potato	Decreased swelling power, amylose leaching	[50]
HMT (100 °C, 16 h, 20, 25, and 30% moisture)	Rice starch and flour	Decreased the gel hardness (gel hardness of flour < starch)	[51]
HMT (100 °C, 10 h, 20 and 25% moisture)	Sorghum starch and flour	Decreased swelling power and solubility. Significant changes in the functional properties of flour > starch	[44]
HMT (110 °C, 16 h, 20, 25, 30, 35% moisture)	Tartary buckwheat starch	Decreased swelling power, oil absorption capacity, and solubility Increased water absorption capacity Decreased adhesiveness, gumminess, chewiness, hardness, and cohesiveness	[52]
HMT (110 °C, 12 h, 15, 20, 25, 30, 35% moisture)	Sweet potato	Decreased swelling power and solubility Decreased the transmission and amylose content of starch/flour	[53]
HMT (90, 100, 110, 120 °C, 20 min)	Sweet potato flour	Significantly increased WI, gas retention, and bread volume	[54]
HMT (85 and 120 °C, 6 h)	Buckwheat	Decreased swelling power, oil absorption capacity, and solubility Increased water absorption capacity and hardness Increased adhesiveness, gumminess, and chewiness, but decreased springiness and cohesiveness	[55]
RHMT (2 cycles, 100 & 120 °C, 2 h)	Rice, cassava, Pinhão bean	Decreased swelling power (cassava > rice and Pinhão bean) Decreased solubility (cassava > Pinhão bean > rice)	[36]
RHMT (5 cycles, 100 °C, 6 h, 30% moisture)	Sweet potato	Decreased swelling power and solubility	[26]
RHMT (120 °C, 2 h) and CHMT (120 °C, 4, 6, 8, 10, 12 h)	Red adzuki bean starch	Decreased swelling power and solubility	[56]
RHMT (6 cycles) and CHMT	Mung beans	Decreased swelling power Swelling power and solubility CHMT > RHMT	[57]
ANN (50 °C, 72 h)	Barley	Decreased swelling power	[46]
RANN (2 cycles, 45, 50, 55, 60 °C, 72 h)	Jackfruit seed starch	Decreased swelling power and solubility Increased gel hardness	[58]
RANN (55 °C; 12 h, 8 cycles) and CANN (55 °C; 96 h)	Potato	Decreased the swelling power and solubility for RANN and CANN	[35]
RANN (55 °C; 12 h, 8 cycles) and CANN (55 °C; 96 h)	Red adzuki bean	Decreased swelling power, solubility, and WHC RANN is more effective than CANN	[59]
RANN (50 °C; 12 h, 8 cycles) and CANN (50 °C; 96 h)	Mung beans	Swelling power and solubility RANN < CANN RANN is more effective than CANN	[34]
RANN (65 °C; 12 h, 8 cycles) and CANN (65 °C; 96 h)	Sweet potato starch	Decreased swelling power and solubility RANN is more effective than CANN	[27]
HMT (100 °C, 16 h, 20%, 25%, 30% moisture).	Finger millet	Increased swelling power and solubility	[60]
ANN (50 °C, 48 h).			
HMT (100 °C, 16 h)	New cocoyam	Decreased the swelling power, solubility, and water absorption capacity	[61]
ANN (50 °C, 24 h)			
HMT (100 or 120 °C, 2 h)	Corn, peas, and lentils	Decreased swelling power	[40]
ANN (10-15 °C < T_o_, 24 h)			
HMT (110 °C, 16 h)	Common buckwheat starch	Decreased swelling power and solubility	[62]
ANN (50 °C, 24 h)			
HMT (100 °C, 6 h)	Sweet potato (white, yellow, and purple)	Decreased swelling power and solubility for ANN	[25]
ANN (45 °C, 24 h)			
HMT (120 °C, 2 h)	Seeds of grass pea	Decreased swelling and amylose leaching	[20]
ANN (10 °C < T_o_, 24 h)			
HMT (100 °C, 8 h)	Banana flour	Decreased swelling power, solubility, freeze-thaw stability, and water absorption capacity except for dual retrogradation (DR)	[63]
ANN (55 °C, 12 h)			
DR (100 °C, 30 min, and 4 °C, 48 h)			
Extrusion + HMT (EHMT) (100 °C, 6 h)	Corn	Decreased swelling power and solubility	[64]

**Table 2 polymers-14-05447-t002:** Several studies related to the effect of hydrothermal modifications on the morphology of starch and flour.

Modification Types	Source of Starch/Flour	Effect on Morphological Properties	References
HMT (100 °C, 10 h, 30% moisture)	Potato, taro, new cocoyam, true yam, cassava	There were no significant changes	[47]
HMT (100 °C, 16 h, 27% moisture)	Sweet potatoes, Peruvian carrots	There were a few cracks on the surface of the granules	[80]
HMT (110 °C, 1 h, 15%, 20%, 25% moisture)	Rice	Granules were more aggregated with a more irregular surface	[28]
HMT (100 °C, 16 h, 25% moisture)	Rice, corn, potato	There was no significant change in corn and potato starch granules There was partial gelatinization of rice starch granules	[81]
HMT (120 °C, 12 h, 15, 20, 25, 30, and 35% moisture)	Mung bean	There were a few cavities on the surface of the starch granules There were differences in the birefringence patterns in the form of decreased, lost, and high birefringence patterns in the center of granules	[48]
HMT (100 °C, 10 h, 20 and 25% moisture)	Sorghum starch and flour	The surface of the sorghum starch granules was more perforated with a regular and small structure The surface of the starch granules of sorghum flour became more regular and had many small holes	[44]
HMT (110 °C, 16 h, 20, 25, 30, 35% moisture)	Tartary buckwheat starch (TBS)	There were holes, cracks, and the more compact the amorphous portion of the starch granules	[52]
HMT (90, 100, 110, 120 °C, 20 min)	Sweet potato flour	The surface of the starch granules was slightly broken, reducing the particle size with a more compact arrangement	[54]
HMT (85 and 120 °C, 6 h)	Buckwheat	There were no significant changes	[55]
RHMT (5 cycles, 100 °C, 6 h, 30% moisture)	Sweet potato	There was agglomeration, enlargement of the pore size, and the shallower and smoother surface of the starch granules	[26]
RHMT (120 °C, 2 h) and CHMT (120 °C, 4–12 h)	Red adzuki bean starch	Starch granules were broken, and there were holes	[56]
RHMT (6 cycles) and CHMT	Mung beans	RHMT starch granule morphology was more regular than CHMT	[57]
ANN (45, 50, 55, 60 °C, 72 h, Single-ANN, Double-ANN)	Jackfruit seed starch	There were no significant changes	[58]
RANN (55 °C; 12 h, 8 cycles) and CANN (55 °C; 96 h)	Potato	There were no significant changes	[35]
RANN (55 °C; 12 h, 8 cycles) and CANN (55 °C; 96 h)	Red adzuki bean	There were no significant changes	[59]
RANN (50 °C; 12 h, 8 cycles) and CANN (50 °C; 96 h)	Mung beans	The surface grooves were deeper, and there were cracks in the center of the starch granules	[34]
HMT (100 °C, 16 h)	New cocoyam (talas Belitung)	There were no significant changes	[61]
ANN (50 °C, 24 h)			
HMT (100 or 120 °C, 2 h)	Corn, peas, lentils	Birefringence in ANN did not change, whereas birefringence in HMT decreased	[40]
ANN (10-15 °C under T_o_, 24 h)			
HMT (110 °C, 16 h)	Common buckwheat starch	There were cavities, gaps, and holes	[62]
ANN (50 °C, 24 h)			
HMT (100 °C, 6 h)	Sweet potato	There were no significant changes	[25]
ANN (45 °C, 24 h)			
HMT (120 °C, 2 h)	Seeds of grass pea	There were no significant changes There was a cavity in the center of the starch granules in HMT	[20]
ANN (10 °C under T_o_, 24 h)			
HMT (100 °C, 8 h)	Banana flour	HMT: cohesive structure with a less compact surface ANN: There were no significant changes DR: there was a cohesive structure with many irregular aggregates	[63]
ANN (55 °C, 12 h)			
DR (100 °C, 30 min, and 4 °C, 48 h)			
HMT 100 °C, 1 h, 22% moisture)	Carioca bean starch	There were shallow indentations and grooves on the surface of the starch granules and an increase in the size of the starch granules in all treatments	[82]
ANN (50 °C, 16 h)			
SNT (ultrasonic processor)			
Extrusion + HMT (EHMT) (100 °C, 6 h)	Corn	The starch granule surface was smoother, and the crystalline portion was relatively larger	[64]

**Table 3 polymers-14-05447-t003:** Several studies related to the effect of hydrothermal modifications on the pasting properties of starch and flour.

Modification Types	Source of Starch/Flour	Effect on Pasting Properties	References
HMT (120 °C, 1 h)	Corn and potato	The degree of retrogradation did not change significantly	[97]
HMT (100 °C, 16 h, 27% moisture)	Sweet potato, Peruvian carrot, ginger	In sweet potato and ginger, a decrease in viscosity, no BD, and a decrease in SB Changes in pasting properties of Peruvian carrot and sweet potato starches were more significant than that of ginger	[80]
HMT (110 °C, 1 h, 15%, 20%, 25% moisture)	Rice	Decreased the PV, BD, FV, and SB, and the most significant changes in pasting properties in high-amylose starch	[28]
HMT (100 °C, 16 h, 25% moisture)	Rice, corn, potato	Decreased viscosity	[81]
HMT (90, 110, 130 °C, 1, 8, 15 h, 18, 23, 28% moisture)	Sweet potato flour	Decreased viscosity and loss of BD	[49]
HMT (120 °C, 3 h, 15, 20, 25% moisture, pH 5.0 or 6.0)	Potato	Slower gelatinization process and decreased viscosity	[50]
HMT (100 °C, 16 h, 20, 25, and 30% moisture)	Rice starch and flour	Decreased the PV, BD, and SB (no significant increase or change in the curve) HMT had a more significant effect on the viscosity of flour than starch	[51]
HMT (100 °C, 10 h, 20 and 25% moisture)	Sorghum starch and flour	Decreased the PV, TV, BD, FV, and SB	[44]
HMT (110 °C, 16 h, 20–35% moisture)	Tartary buckwheat starch	Decreased the PV, BD, and SB	[52]
HMT (110 °C, 12 h, 15–35% moisture)	Sweet potato starch	Decreased the PV, BD, and easy to retrograde	[53]
HMT (85 and 120 °C, 6 h)	Buckwheat	Decreased the PV, BD, FV, and SB	[55]
HMT (100 °C & 120 °C, 2 h, 2 cycles)	Rice, cassava, Pinhão seeds	Decreased the PV and BD SB: rice starch > Pinhão seeds and cassava FV: rice starch > Pinhão seeds > cassava	[36]
RHMT (5 cycles, 100 °C, 6 h, 30% moisture)	Sweet potato starch	Decreased the overall viscosity, and loss of BD (starting at RHMT-2)	[26]
RHMT (120 °C, 2 h) and CHMT (120 °C, 4–12 h)	Red adzuki bean starch	Decreased the PV, BD, FV, and SB	[56]
RHMT (6 cycles) and CHMT	Mung beans	Decreased the PV, TV, BD, FV, and SB on RHMT (until RHM4) and CHMT (CHMT8) The viscosity of RHM < CHM Pasting properties with the best heat stability was shown by HMT four cycles	[57]
ANN (45–60 °C, 72 h, Single-ANN, Double-ANN)	Jackfruit seed starch	PV: Double-ANN < NS < Single-ANN Retrogradation tendency: single-stage annealed > double-stage annealed	[58]
RANN (55 °C; 12 h, 8 cycles) and CANN (55 °C; 96 h)	Potato	Decreased the BD RANN was more effective than CANN	[35]
RANN (55 °C; 12 h, 8 cycles) and CANN (55 °C; 96 h)	Red adzuki bean	Decreased the PV, BD, FV, and SB as cycles and modification time increased	[59]
RANN (50 °C; 12 h, 8 cycles) and CANN (50 °C; 96 h)	Mung beans	Decreased the PV and BD Increased the TV, FV, SB, and peak time on RANN and CANN PV and BD RANN < CANN, FV and BD RANN > CANN on 4 cycles (48 h)	[34]
RANN (65 °C; 12 h, 8 cycles) and CANN (65 °C; 96 h)	Sweet potato starch	Increased the TV, FV, and SB but decreased the PV and BD RANN was more effective in up to 4 cycles in modifying pasting properties than CANN	[27]
HMT (100 °C, 16 h, 20%, 25%, 30% moisture).	Finger millet	Decrease in PV, BD, FV, and TV, but an increase in TV	[60]
ANN (50 °C, 48 h)			
HMT (100 °C, 16 h)	New cocoyam (talas Belitung)	Decreased the PV, HV, FV, SB, BD, and improved heat and pressure stability	[61]
ANN (50 °C, 24 h)			
HMT (110 °C, 16 h)	Common buckwheat starch	Decreased the PV, BD, and SB along with the increase in moisture	[62]
ANN (50 °C, 24 h)			
HMT (100 °C, 6 h)	Sweet potato	HMT: decreased the PV, BD, and SB as well as the increase in TV and FV ANN: slightly decreased in PV and BD	[25]
ANN (45 °C, 24 h)			
HMT (110 °C, 24 h)	Acorn	HMT and ANN: a significant increase in peak time, a significant decrease in PV, BD, TV, FV, and SB, as well as a decrease in the tendency of retrogradation, but the effect of HMT was more significant Dual modification: HMT-ANN had a more significant impact than ANN-HMT	[17]
ANN (50 °C, 24 h)			
Dual modification (HMT-ANN, ANN-HMT)			
HMT (120 °C, 2 h)	Seeds of grass pea	Modification and storage conditions had not been effective in changing the tendency of starch retrogradation (HMT)	[20]
ANN (10 °C under T_o_, 24 h)			
HMT (100 °C, 8 h)	Banana flour	Decreased the PV and BD of all samples and decreased in HV, FV, and SB of all samples except for the ANN modification HMT had a more significant impact than ANN and DR on pasting properties	[63]
ANN (55 °C, 12 h)			
DR (100 °C, 30 min, and 4 °C, 48 h)			
HMT 100 °C, 1 h, 22% moisture)	Carioca bean starch	The combination of modified SNT increased PV (co: ANN-SNT, SNT-ANN), while HMT decreased viscosity, especially BD, and effectively increased heat stability and decreased setback compared to ANN	[82]
ANN (50 °C, 16 h)			
SNT (ultrasonic processor)			

**Table 4 polymers-14-05447-t004:** Several studies related to the effect of hydrothermal modifications on the crystallinity of starch and flour.

Modification Types	Source of Starch/Flour	Effect on Crystallinity	References
HMT (120 °C, 1 h)	Corn, potato	Decrease in crystallinity index, especially in potato starch	[97]
HMT (100 °C, 10 h, 30% moisture)	Potato, taro, new cocoyam, true yam, cassava	Decrease in crystallinity index with changes in the crystallinity type of potato and true yam (type B → type C (A + B)) Decrease in crystallinity index without change in new cocoyam and taro (type A) and cassava (type C (A + B))	[47]
HMT (100 °C, 16 h, 27% moisture)	Sweet potato, Peruvian carrot, ginger	Decrease in crystallinity index of the Peruvian carrot with a change in the type of crystallinity (type B to type A) Slight decrease in crystallinity index without changes in the type of crystallinity of sweet potato and ginger (still type A)	[80]
HMT (110 °C, 1 h, 15%, 20%, 25% moisture)	Rice	Decrease in the relative crystallinity index	[28]
HMT (100 °C, 16 h, 25% moisture)	Rice, corn, potato	Decrease in the crystallinity index of all samples except potatoes (waxy) The crystallinity type of rice and corn starch remained type A, while potato starch type B became type A	[81]
HMT (120 °C, 12 h, 15–35% moisture)	Mung beans	An increase in the crystallinity index without a change in the crystallinity type (type A)	[48]
HMT (90, 110, 130 °C, 1, 8, 15 h, 18, 23, 28% moisture)	Sweet potato flour	Decrease in crystallinity index without a change in crystallinity type (type A)	[49]
HMT (120 °C, 3 h, 15, 20, 25% moisture, pH 5 or 6)	Potato	Increase in crystallinity index	[50]
HMT (100 °C, 10 h, 20, 25% moisture)	Sorghum starch and flour	Increase in crystallinity index without change in crystallinity type (type A)	[44]
HMT (110 °C, 16 h, 20–35% moisture)	Tartary buckwheat starch	Increase in crystallinity index without change in crystallinity type (remains to type A)	[52]
HMT (85 °C and 120 °C, 6 h)	Buckwheat	Decrease in crystallinity index without changing the crystallinity type (remains to type A)	[55]
Single HMT (100 °C & 120 °C, 2 h)	Rice, cassava, Pinhão seed	Decrease in crystallinity index in both cycles with changes in the type of crystallinity of Pinhão seeds (type C to type A) and without changes in rice and cassava (type A)	[36]
Double HMT (100 °C & 120 °C, 2 h, 2)			
RHMT (100 °C, 6 h, 30% moisture, 5 cycles)	Sweet potato starch	Decrease in crystallinity index with a change in crystallinity type (CA type to type A) The effect of the number of cycles is more significant than time	[26]
RHMT (120 °C, 2 h) and CHMT (120 °C, 4, 6, 8, 10 and 12 h)	Red adzuki bean starch	Decrease in crystallinity index and changes in crystallinity type (CHMT and RHMT from type C to type A). Loss of polarization cross	[56]
RHMT (6 cycles) and CHMT	Mung beans	Increase in crystallinity index without changing the crystallinity type of CHMT and RHMT (remained type A). Degree of crystallinity RHMT > CHMT	[57]
ANN (50 °C, 72 h)	Barley	Increase in the crystallinity index of waxy and normal starches without a change in the type of crystallinity (remains to type A) There was no change in the crystallinity index of high amylose starch accompanied by a change in the type of polymorph (type C (A + B) to type A)	[46]
Cycles ANN (45–60 °C, 72 h, Single-ANN, Double-ANN)	Jackfruit seed starch	Increase in crystallinity index without changing the crystallinity type (type A). The crystallinity of single-ANN > double-ANN	[58]
Cycles RANN (55 °C; 12 h, 8 cycles) and CANN (55 °C; 96 h)	Potato	Increase in crystallinity index without changing the crystallinity type (type B) in RANN and CANN. The crystallinity of RANN > CANN	[35]
Cycles RANN (55 °C; 12 h, 8 cycles) and CANN (55 °C; 96 h)	Red adzuki bean	Increase in crystallinity index without changing the crystallinity type (type C) in RANN and CANN	[59]
Cycles RANN (50 °C; 12 h, 8 cycles) and CANN (50 °C; 96 h)	Mung beans	Increase in crystallinity index without changing the type of crystallinity (type C) in RANN and CANN. The crystallinity index of RANN > CANN	[34]
Cycles RANN (65 °C; 12 h, 8 cycles) and CANN (65 °C; 96 h)	Sweet potato starch	Increase in crystallinity index without changing the crystallinity type (type A) of RANN and CANN	[27]
HMT (100 °C, 16 h, 20%, 25%, 30% moisture). ANN (50 °C, 48 h)	Finger millet	Slight decrease in crystallinity index with a change in the type of crystallinity in HMT (type C to A) Slight decrease in crystallinity index with a change in crystallinity type in ANN (type B to A)	[60]
ANN (50 °C, 24 h)	New cocoyam	There was no change in the type of crystallinity in HMT and ANN (type A)	[61]
HMT (100 °C, 16 h)			
HMT (90–130 °C, 24 h, 17, 20, 23, 26% moisture)	Potato	Decrease in crystallinity index and change in crystallinity type (type B to A)	[103]
ANN (50 °C, 24 h)	Common buckwheat starch	Increased RC in ANN and HMT without changing crystallinity type (type A)	[62]
HMT (110 °C, 16 h)			
HMT (100 °C, 6 h)	Sweet potato	Increase in the crystallinity index of HMT without a change in the crystallinity type (type A) Decrease in crystallinity index of ANN without change in crystallinity type (type A)	[25]
ANN (45 °C, 24 h)			
ANN (10 °C under T_o_, 24 h)	Seeds of grass pea	Increase in the crystallinity index without changing the type of crystallinity in the ANN (remained in type C), while the HMT changed from type C to A HMT crystallinity index > ANN, so HMT was more compact.	[20]
HMT (120 °C, 2 h)			
HMT (100 °C, 8 h)	Banana flour	Increase in crystallinity index in HMT and ANN but decrease in crystallinity index in DR Change in the crystallinity type of HMT (type B to A) and DR (type B to A + B) There was no change in the type of crystallinity of ANN and the formation of type V crystals in all modified flour	[63]
ANN (55 °C, 12 h)			
DR (100 °C, 30 min, and 4 °C, 48 h)			
Ekstrusi-HMT (EHMT) (100 °C, 6 h)	Corn	Increase in crystallinity index and change in crystallinity type (type V to V–A)	[64]

**Table 5 polymers-14-05447-t005:** Several studies related to the effect of hydrothermal modifications on thermal properties.

Treatment	Effect on Thermal Properties	References
Single HMT and cycles	Increase in gelatinization temperature	[17,20,26,28,36,40,47,49,51,52,55,56,57,60,62,80,81,82]
Single ANN and cycles	Increase in gelatinization temperature	[17,40,46,60,62,82,103]
ANN	Decrease in gelatinization temperature	[20]
Single HMT and cycles	Decrease in heating enthalpy	[17,26,40,47,49,52,54,55,62,81,82,112]
ANN	Decrease in heating enthalpy	[40,46,62]
Single HMT and cycles	Increase in heating enthalpy	[44,61]
Single ANN and cycles	Increase in heating enthalpy	[27,35,59,61]
ANN	No significant change in heating enthalpy	[17]
HMT	Increase in the gelatinization temperature range	[40,52,56,57]
Single HMT and cycles	Decrease in the gelatinization temperature range	[17,26,49]
Single ANN and cycles	Decrease in the gelatinization temperature range	[17,27,40,46]

**Table 6 polymers-14-05447-t006:** Some applications of hydrothermal-treated starch and flour (HMT and ANN) for food products.

Starch and Treatment	Product	Result	References
HMT-amaranth starch	Amaranth noodle	HMT-amaranth starch improved the quality of noodles with minimum cooking time, had a good expansion and firm texture	[117]
HMT-potato starch	Tomato sauce	The addition of 1.5% HMT-potato starch improves the stability of tomato sauce during heating	[38]
HMT-wheat and barley flours	Bread	The use of HMT-wheat and barley flours produced the same bread properties and bio-accessibility as untreated flours	[121]
HMT-sweet potato starch	Edible film	The addition of 1.5% HMT-sweet potato starch improved the physicochemical properties, where the elongation, tensile strength, and thickness increased, but the solubility decreased	[119]
HMT/oxidized-potato starch	Biofilms	HMT-potato starch increased the tensile strength and decreased the water vapor permeability of potato starch films	[120]
HMT- and ANN-rice starch	Rice noodle	The substitution of 50% HMT- or ANN-rice starch improved the cooking and texture quality of rice noodle	[65]
ANN-rice starch	Rice noodle	The substitution of 40% ANN-rice starch improved the cooking quality and sensorial properties of rice noodles	[118]
ANN-waxy rice starch	Starch nanoparticles	ANN-waxy rice starch nanoparticles had crystallinity and melting temperature higher than untreated starch	[122]

## Data Availability

Not applicable.

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
