# Peer review of "A Mini Review of Physicochemical Properties of Starch and Flour by Using Hydrothermal Treatment"

_polymers, 2022, doi:10.3390/polym14245447_

Round 1

Reviewer 1 Report

A very comprehensive approach to discuss modification of flour and starch by hydrothermal treatments. The authors made a comprehensive comparison of the functional, pasting, and physicochemical properties of hydrothermally modified starch and flour (ANN and HMT) from various plant sources using available publications.

Personally, it is better to read a publication that presents graphs, diagrams, photographs. There is no graphical representation of the results, which would certainly be an added value.

There are mentions of SEM, LM, PLM, and CLSM analysis, it begs to show such examples and cite sources.

It would be nice to see graphic interpretations, but despite its lack, the subject was treated properly.

Author Response

Response to Reviewer 1 Comments

A very comprehensive approach to discuss modification of flour and starch by hydrothermal treatments. The authors made a comprehensive comparison of the functional, pasting, and physicochemical properties of hydrothermally modified starch and flour (ANN and HMT) from various plant sources using available publications.

Point 1: Personally, it is better to read a publication that presents graphs, diagrams, photographs. There is no graphical representation of the results, which would certainly be an added value.

Response 1:

Graphs, diagrams, and photographs have been added to the manuscript to clarify explanations (Figure 1-Figure 6: Page 2, Line 80-84, Pages 9-10, Line 190-204, Page 18, Line 349-357, red color).

Point 2: There are mentions of SEM, LM, PLM, and CLSM analysis, it begs to show such examples and cite sources. 

Response 2:

The examples of SEM, LM, PLM, and CLSM analysis have been added to the manuscript (Figure 2, Figure 3, and Figure 4: Pages 9-10, lines 190-204, red color).

Point 3: It would be nice to see graphic interpretations, but despite its lack, the subject was treated properly.

Response 3:

The graphic interpretations have been added to the manuscript (Page 17, lines 329-341, and Page 18, lines 349-357, red color).

Reviewer 2 Report

This review is clear, easy to follow and to understand, well-written, and well-organized. Many references are from 2021 and 2022. Present drawbacks and future research are identified. These five Tables seem good to me, although, depending on the final pagination, it may be necessary to put some "cont." headers. I guess this can be done during proofing, should the article be accepted.

Some issues:

I am troubled by the lack of figures, since the review actually has some opportunities for them. For instance, depicting the granular structure of starch, amylose, amylopectin, type A, type B... The text has claims about granules, macromolecules, and supramolecular arrangements that could be, besides explained, visualized.

Speaking of which, line 83: "starch granule molecules" -> "starch molecules". Indeed, these are the molecules of/in the starch granule, so it was not wrong — just ambiguous. 

Likewise, since X-ray diffraction is mentioned, the authors could grab some pattern from the literature (with due permission!), or from their own files, to illustrate how it is and how it changes when starch loses crystallinity. By the way, please capitalize the "X" (line 272).

Line 123: "an increase in amylose-amylopectin chain interactions that strengthen hydrogen bonds". Is this not redundant? As far as I know, those chain interactions are mainly hydrogen bonds.

To my judgment, this overview should be published once a couple figures are incorporated. I suggest ChemBioDraw, Avogadro or any other software of choice to depict molecular structures.

Author Response

Response to Reviewer 2 Comments

This review is clear, easy to follow and to understand, well-written, and well-organized. Many references are from 2021 and 2022. Present drawbacks and future research are identified. These five Tables seem good to me, although, depending on the final pagination, it may be necessary to put some "cont." headers. I guess this can be done during proofing, should the article be accepted.

Point 1: I am troubled by the lack of figures, since the review actually has some opportunities for them. For instance, depicting the granular structure of starch, amylose, amylopectin, type A, type B... The text has claims about granules, macromolecules, and supramolecular arrangements that could be, besides explained, visualized.

Response 1:

The figures about the granular structure of starch, amylose and amylopectin, and the type of starch (type A, type B, and type C) have been added for visualization (Figure 2 - Figure 4: Pages 9-10, lines 190-204, Figure 1: page 2, Figure 5-6, Lines 80-84, Pages 18, lines 349-357, red color).

Point 2: Speaking of which, line 83: "starch granule molecules" -> "starch molecules". Indeed, these are the molecules of/in the starch granule, so it was not wrong — just ambiguous. 

Response 2:

"starch granule molecules" has been revised to "starch molecules" to make it not ambiguous (Page 5, Line 103, red color).

Point 3: Likewise, since X-ray diffraction is mentioned, the authors could grab some pattern from the literature (with due permission!), or from their own files, to illustrate how it is and how it changes when starch loses crystallinity. By the way, please capitalize the "X" (line 272).

Response 3:

The patterns of X-ray diffraction of starch have been added to the manuscript to illustrate how it is and how it changes when starch loses crystallinity (Page 18, lines 349-357, red color).

Point 4: Line 123: "an increase in amylose-amylopectin chain interactions that strengthen hydrogen bonds". Is this not redundant? As far as I know, those chain interactions are mainly hydrogen bonds.

Response 4:

The sentence "an increase in amylose-amylopectin chain interactions that strengthen hydrogen bonds" has been revised to that those chain interactions are mainly hydrogen bonds (Page 6, Lines 142-145, red color).

Point 5: To my judgment, this overview should be published once a couple figures are incorporated. I suggest ChemBioDraw, Avogadro or any other software of choice to depict molecular structures.

Response 5:

The figures have been incorporated, and molecular structures have been added to the manuscript (Page 2, Line 80-84, Pages 9-10, Line 190-204, Page 18, Line 349-357, red color).

Author Response

Response to Reviewer 3 Comments

This is an interesting manuscript describing the “Hydrothermal Treatment for The Modification of Starch and Flour: A Mini Review of Impact on Physicochemical Properties”. The paper describes the a comprehensive comparison of the functional, pasting, and physicochemical properties of hydrothermally modified starch and flour using annealing and heat moisture treatment from various plant sources so that they can be used as a reference for the development of modified starch and flour for various industrial needs.

This review is interesting and helpful for readers in food science and biotechnology industry. The authors are suggested to perform major revisions of the manuscript on the following points given below:

Point 1: The title should be “A mini review of physicochemical properties of Starch and Flour by using hydrothermal treatment”

Response 1:

The title has been revised to "A Mini Review of Physicochemical Properties of Starch and Flour by Using Hydrothermal Treatment" (Page 1, Line 2-3).

Point 2: Authors should re-write the introduction part mentioning the advantages of hydrothermal treatment and highlighted in the revised manuscript.

Response 2:

The advantages of hydrothermal treatment have been added in the introduction part (Page 1, Line 30-35, red color).

Point 3: Authors should mention the comparative study of different treatments over hydrothermal.

Response 3:

The comparative study of different treatments over hydrothermal has been mentioned (Page 1, Line 36-43, red color).

Point 4: The authors should add the morphological view images of HMT and ANN of various hydrothermal treatments of Starch and Flour.

Response 4:

The morphological view images of HMT and ANN of hydrothermal treatments of Starch and Flour have been added (Figure 2, Figure 3, Figure 4: Pages 9-10, lines 190-204, red color).

Point 5: Can authors add some applications of hydrothermal treated starch and flour?

Response 5:

Some applications of hydrothermal treated starch and flour have been added (Page 20-21, Line 401-415, red color).

Point 6: Is HMT has greater effect on the solubility and crystallinity of flour than of starch?

Response 6:

Generally, HMT has a greater effect on the solubility and crystallinity of flour than on starch. Still, in certain starches, the effect can be different, which is affected by the amylose-amylopectin composition in starch and non-starch components in flours. An explanation about this has been added to the manuscript (Page 3, Line 85-92, red color).

Round 2

Reviewer 3 Report

Please, accept the revision. Corrected according to comments. Thanks